# Pentagonal photonic crystal mirrors: scalable lightsails with enhanced acceleration via neural topology optimization

Lucas Norder [1], Shunyu Yin[2], Matthijs H. J. de Jong [1,3], Francesco Stallone[4], Hande Aydogmus[4], Paolo M. Sberna[5], Miguel A. Bessa [2] ✉ & Richard A. Norte [1,3] ✉

The Starshot Breakthrough Initiative aims to send gram-scale microchip probes to Alpha Centauri within 20 years, propelled by laser-driven lightsails at a fifth of light speed. This mission demands innovative lightsail materials with meter-scale dimensions, nanoscale thickness, and billions of nanoscale holes for enhanced reflectivity and reduced mass. Unlike the microchip payload, lightsail fabrication requires breakthroughs in optics, materials science, and structural engineering. Our study uses neural topology optimization, revealing a novel pentagonal lattice-based photonic crystal (PhC) reflector. The optimized designs significantly lower the acceleration times and, thereby, launch cost. Crucially, they also enabled orders-of-magnitude fabrication cost reduction. We fabricated a $60 \times 60$ mm$^2$, 200 nm thick reflector with over a billion nanoscale features, achieving a 9000-fold cost reduction per m$^2$. This represents the highest aspect ratio nanophotonic element to date. While stringent requirements remain for lightsails, scalable, cost-effective nanophotonics present promising solutions for next-generation space exploration.

Currently, the human-made object furthest from Earth is the Voyager 1[1]. Traversing space since 1977, this spacecraft has only recently left our solar system, a mere 0.5% of the distance to the nearest star outside our solar system; Alpha Centauri. With existing propulsion systems, approaching our nearest interstellar neighbor would take over 10,000 years. In 2016, the Breakthrough Prize Foundation announced the Starshot Initiative to push the development of low-mass microchip satellites with cameras, sensors, and probes accelerated to high speeds by low-mass lightsails[2]. The Starshot Mission leverages advances in nanotechnology to create low-mass objects, and achieve progress in high-power lasers that beam energy to distant locations as far as tens of millions of kilometers away[3].

This microchip approach to space exploration aims to reach Alpha Centauri (i.e., the nearest star outside our Solar system) within 20 years by reaching speeds up to 20% of the speed of light, made possible by the rapidly advancing field of nanotechnology and the future potential of next-generation laser systems. Regardless of the payload mass, this mission is fundamentally exploring the physical limits of mass acceleration and our ability to reach relativistic speeds with novel materials made possible by nanotechnology. Of the many ambitious developments required by the Starshot Initative, the lightsails are generally considered one of the most challenging components to realize due to their unique geometries and stringent performance requirements.

[1]Department of Precision and Microsystems Engineering, Delft University of Technology, Delft, The Netherlands. [2]School of Engineering, Brown University, Providence, RI, USA. [3]Department of Quantum Nanoscience, Kavli Institute of Nanoscience, Delft University of Technology, Delft, The Netherlands. [4]Else Kooi Lab, Delft University of Technology, CT Delft, The Netherlands. [5]Department of Microelectronics, THz Sensing group, Delft University of Technology, Delft, The Netherlands. ✉e-mail: miguel_bessa@brown.edu; r.a.norte@tudelft.nl

The Starshot concept, presented in Fig. 1, is based on generating an optical force on a reflective lightweight sail material by projecting a high-power Earth-based laser on it. As proposed in the Starshot initiative, the lightsail will be ~10 m² and the laser power 10–100 GW/m² to generate sufficient radiation pressure[2] within a few minutes of laser exposure. From Eq. (1), it follows that in this short exposure time, the maximum radiation pressure on the sail using a 1 GW/m² laser is around 6.7 N/m².

$$p_r = \frac{2I}{c} \tag{1}$$

To approach relativistic speeds (0.2c), stringent low-mass budgets are required, limiting the weight of the sail and the connected payload chip to approximately 1 gram each. The laser used for radiation pressure on the sail needs to operate on wavelengths in the near-infrared (NIR) spectrum from 1 to 2 µm because of its low atmospheric absorption[4]. These lightsails will experience Doppler-shifts as they accelerate, requiring high broadband reflectivity[5]. Larger bandwidth can generally be achieved by increasing the thickness of the sails at the cost of additional mass, which can severely reduce its acceleration performance. Given the interaction with a high-power laser beam, lightsails must achieve ultra-low optical absorption to avoid thermal fracturing. While all components from payload to lightsails will require significant development over the next decades, the lightsail stands out as the major challenge of this initiative because of its unique requirements. Achieving a 1 g microchip payload will require miniaturizing all of its components like cameras, communications, and sensors in x, y and z dimensions. On the contrary, achieving gram-scale, 10 m² lightsails will require spanning a reflector to meter scales in x and y while retaining nanoscale thickness – far from any aspect-ratio achievable by modern nanotechnology. The physics and economics of how these high-aspect ratio reflectors are made will be crucial to the success of this technology.

One often neglected aspect of this mission is that these long-distance missions rely on a shotgun approach of many sails to increase the chance of success. This means the costs of manufacturing and launching these sails with high power (for several minutes) are major considerations that have not been taken into account in the design process of the sails but are crucial to Starshot's ambitious goals.

Many possible lightsail materials are proposed in the literature[5–11]. Among these materials, single-layer silicon nitride (SiN) photonic crystals are the top candidate material because SiN combines low optical absorption and the low mass and high reflectivity achieved by single-layer hole-based photonic crystals. Advantageously, silicon nitride is a well-studied and mature CMOS material that can be conventionally integrated with many microchip platforms. Photonic crystals made from SiN have been well studied in the field of optomechanics, which also favors low mass and absorption with high reflectivity[12–18]. Additionally, SiN membranes will not wrinkle due to the internal tensile stresses generated in the deposition allowing for better stability once suspended. This pre-stress in SiN photonic crystals allows for precise alignment of optical beams onto the suspended photonic crystals in a lab-scale test setup. Due to these favorable properties, SiN is chosen as the lightsail material for this work.

Given that photonic crystal reflectors rely on a two-dimensional array of subwavelength holes in a single-layer SiN membrane, it is important to note that there is a direct relation between the minimum feature size (MFS) of the patterns (e.g., minimum distance between holes), and the costs of manufacturing the lightsail; lower MFS means higher costs (more intricate geometric details) but potentially lower mass and better acceleration. This sets up a complex trade-off between cost[19], manufacturing and acceleration performance that has not been previously considered. Additionally, a bigger MFS and larger surface

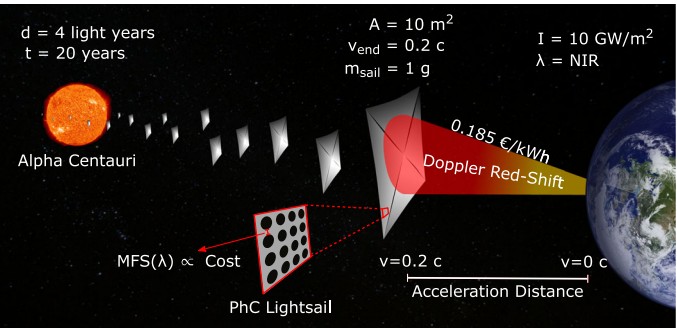

**Fig. 1 | Mission parameters for lightsail mission to Alpha Centauri.** d, distance; t, travel time; A, ligthsail area; v, velocity; m, mass; I, power density; λ, laser wavelength. High power earth-based laser propelling a fleet of lightweight sails to 20% of the speed of light, to reach Alpha Centauri in 20 years[2]. The lightsail needs to be reflective over a broad bandwidth due to the Doppler red-shift of the laser resulting from the change in velocity of the sail. The minimum feature size (MFS) of a photonic crystal based lightsail is related to the fabrication cost. A commonly used performance metric for a lightsail is the acceleration distance. The launch cost is mainly determined by the energy consumption of the laser[43].

area of the sail, can be favorable to crucial properties like stress reduction and increased radiative cooling.

Although single-layer photonic crystals have proven to be effective reflectors even with simple two-dimensional hole lattice designs, the few contributions targeting lightsail design have not considered state-of-the-art manufacturing constraints. The traditional optimization of photonic structures is highly iterative and relies on domain knowledge from experienced researchers[20]. This trial-and-error process is unlikely to be successful in finding high-performance designs because of the high-dimensional design space. Additionally, photonics optimization is usually non-convex, resulting in a challenging optimization. Notwithstanding, inverse design methods have resulted in promising, non-trivial and high-performance PhC designs[21–24] even for lightsail design[6].

Recently, a new inverse-design method referred to as neural topology optimization (neural TO) has been proposed where conventional TO is enhanced by machine learning via the reparameterization approach proposed by Hoyer et al.[25]. This strategy differs from most machine learning contributions aimed at improving inverse design methods. Usually, machine learning is used in inverse design by training generative models such as variational autoencoders and generative adversarial neural networks[26–28] that require large training databases and have difficulties with predictions that fall out of the training data distribution. In contrast, neural TO introduces a neural network before a physics solver (e.g., finite element analyses) and shifts the optimization problem to finding the weights and biases of the neural network that minimize the objective function calculated by the physics solver. Neural TO is still in its infancy and has not been applied in the context of inverse problems in photonics. However, we find that the method is particularly advantageous for lightsail design when compared to conventional TO strategies. Additional information regarding the employed optimization algorithm is presented in the supplementary information.

In this work, we design a photonic crystal lightsail that maximizes acceleration capabilities while minimizing mission costs by addressing both the MFS constraints imposed by lithography processing and the costs associated with laser time needed to accelerate the lightsail. We find that optimizing solely for acceleration capabilities can lead to designs with less mass[6] which are more delicate and difficult to fabricate and launch. A key insight is that the costs of manufacturing lightsails is closely tied to the MFS of the photonic crystal – which can be incorporated in our PhC design optimization. Simultaneously achieving state-of-the-art performance and low costs requires

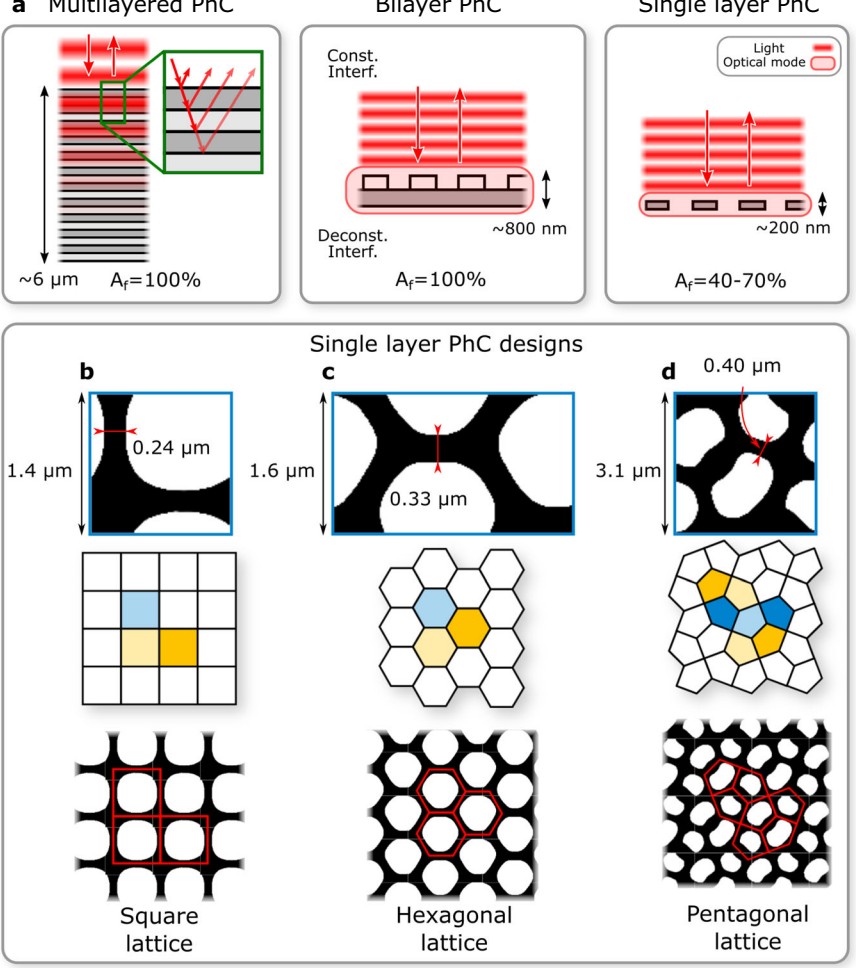

**Fig. 2 | Working principle and designs for photonic crystal reflectors. a** Working principles of different photonic crystal (PhC) architectures. Multilayered PhC consists of stacked layers with varying refractive indices. The bilayer PhC consists of a repeating PhC pattern on top of a solid membrane. Single layer PhC is a membrane with a repetitive PhC hole pattern. For both the bilayer and single-layer PhC, the incident light creates an optical mode within the material that deconstructively interferes with the transmitted light and constructively with the reflected light. The best optimized single layer PhC design without area constraint for square lattice (**b**) and hexagonal lattice (**c**) where black is material and white is vacuum. The square and hexagonal lattice thicknesses are 0.2 μm and 0.3 μm respectively. **d** The pentagonal lattice design for an Area fraction $A_f$ of 55% with a thickness of 0.18 μm.

navigating a complex parameter space. For this, new developments in neural topology optimization are adapted to meet these lightsail design challenges and resulted in the discovery of an unique PhC pentagonal lattice designs. We then show we can produce these wafer-scale lightsail materials at nearly four orders-of-magnitude reduction in manufacturing costs. These reduced costs allow us to reliably produce the highest-aspect-ratio nanophotonic element to date.

## Results

### Photonic crystal lightsail design

The stringent Starshot mission requirements have driven research on free-standing photonic crystals (PhCs) as broadband reflectors. PhCs control light propagation by tuning sub-wavelength variations in refractive index materials[29].

Figure 2a illustrates the working principles of different PhC architectures. The most well-known reflectors for mirror coatings are multilayered photonic crystals, or distributed Bragg reflectors (DBRs), which consist of several layers of dielectric materials with alternating refractive indices and subwavelength thicknesses. These multilayered PhCs, typically several microns thick, can achieve very high reflectivity (>99.5%) over a broad bandwidth (≈200 nm). However, they are too massive for the lightsail requirements (hundreds of grams for a 10 m² sail), and their ultra-high reflectivity is not particularly useful for acceleration.

In contrast, single-layer photonic crystals achieve changes in refractive index through periodic holes in a membrane, providing alternating refractive indices in the x and y directions. Incoming light creates an optical mode in the membrane that constructively interferes with incoming light and destructively interferes with transmitted light, resulting in high reflectivity (≈98.9%) over a narrower range (20 nm). This type of photonic crystal offers an ultra-thin geometry. Due to their design flexibility and single-layer nature, two-dimensional PhCs are expected to offer higher reflectivity for a lower mass[9], as the small film thickness and holes that can reduce nearly half the mass. Single-layer PhCs are currently the only architecture thin enough to achieve a 1-g lightsail. The Area fraction ($A_f$) is the fraction of the area occupied by the material, and single-layer photonic PhCs typically have $A_f$ = 40–70%. Single-layer photonic crystals thus have ultra-low masses, but suffer from narrow-band reflection, limiting it's reflectivity over a large Doppler shift of a lightsail.

Bilayer photonic crystals offer a hybrid approach by increasing thickness (up to a micron)[8,10,30–32] to enhance reflection bandwidth (Fig. 2a), but this trade-off adds mass, significantly impacting the sail's acceleration. Recent efforts have explored a bilayer photonic

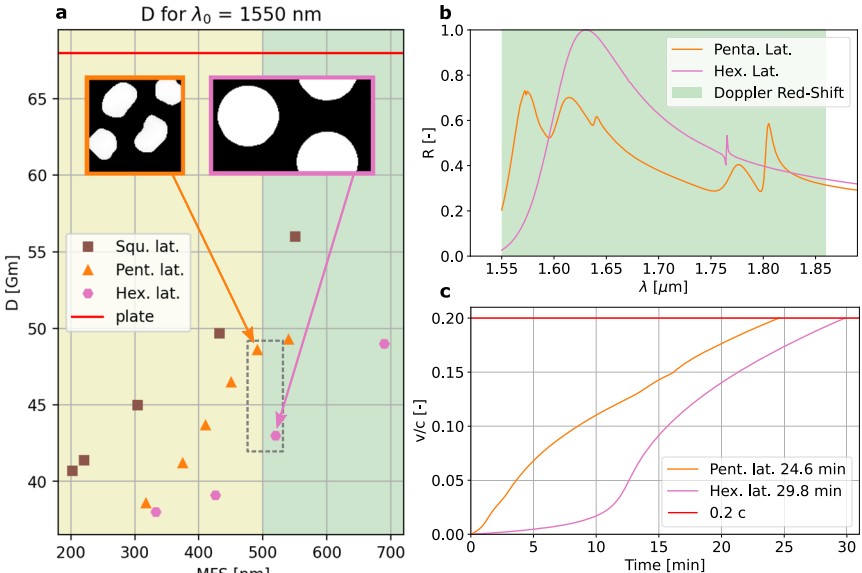

**Fig. 3 | Photonic crystal unitcell optimized for acceleration distance.**
**a** Acceleration distance (*D*) for different lattice structures with varying minimum features size (MFS). The red line indicates the D for a 200 nm thick un-patterned PhC membrane. lat, lattice; Squ., square, Penta., pentagonal; Hex, hexagonal. **b** The

reflectivity spectrum of the selected photonic crystal designs shown in **a** for the full Doppler shift region. The rest of the energy is transmitted due to the ppm absorption of SiN[37]. **c** The velocity of the hexagonal and pentagonal PhC lightsail during acceleration compared with the speed of light.

crystals[30], which consist of one uniform layer and an additional layer of single-layer photonic crystal on top to increase the bandwidth and ease fabrication. However, these types of reflectors can quickly perform worse in terms of acceleration than a single-layer, unpatterned (low-reflectivity) SiN membrane, which serves as our standard worst-case scenario. For instance, bilayer PhCs with a SiN PhC on a Si layer with an average reflectivity of 80% have a similar acceleration distance of 67 Gm compared to a 35% reflective and 200 nm thick unpatterned SiN membrane, as shown in Fig. 3a (red line). This is in agreement with Atwater et al.[5], and highlights the challenging mass requirements for designing lightsails thicker than a single layer to achieve broader bandwidths. Consequently, given the stringent mass requirements, we focus on single-layer PhCs, which are more suitable for lightsail design than other PhC architectures. The supplementary information includes a study discussing the limits of the initially proposed mass target using two-dimensional PhC sails.

### Acceleration performance vs. costs

In designing the lightsail, we must consider not only its acceleration performance but also the associated costs, including those resulting from lithography, manufacturability, and yield, which ultimately impact the final costs. This complexity arises because optimizing for acceleration often leads to designs with low Area fraction and thickness, while manufacturing costs and yield would be significantly reduced with high Area fraction. This sets up a challenging set of tradeoffs in designing and optimizing a photonic crystal that balances both acceleration performance and costs.

In terms of cost, we focus on the lithography process to reduce the fabrication cost as it takes the most time and money compared to other fabrication steps, especially when scaled to square meter-sized PhCs. Selecting a fitting lithography method for patterning the PhC-based lightsail is an integral part of the nano-photonics fabrication due to its direct impact on the achievable resolution and writing speed. E-beam lithography is commonly used for nm-sized structures, yet it is slow and expensive for more extensive areas[33]. E-beam writing times for 1 cm² can vary from multiple days for conventional techniques[9] to numerous hours for the most advanced methods[30,34]. However, a faster and more affordable nanofabrication method is optical lithography[35,36]. Therefore, i-line photolithography (i.e., light

source with 365 nm light), which has a typical MFS of ~500 nm, was selected for this study based on its cost-effectiveness, availability, and established processing protocols. Additionally, the writing time is independent of the design because of the use of a mask, making it a good match for the possible irregular and non-trivial design generated by the neural TO. In the supplementary information, a more detailed comparison is made between the different photolithography methods.

The writing costs are related to the operating costs of a cleanroom, which are expected to be currently around 200 euro/hr. When choosing optical over e-beam lithography, the writing time of a 10 m² sail can markedly be reduced from 15 years to one day, calculated with $7.5 \times 10^{-5}$ m²/h and 0.43 m²/h respectively. Therefore, the cost can be reduced almost 9000 times, from 26 million euro to 3000 euro per sail.

Within the Starshot initiative, there is no agreement yet on which wavelength the laser uses. Specifically for the lightsail development, 1550 nm is the preferred wavelength because the feature size of the PhC is proportional to the wavelength. Therefore, the fabrication cost and complexity are reduced due to the larger features. Furthermore, the optical absorption of SiN is lower for 1550 nm light[16,37], allowing for the use of high-power lasers. As an additional benefit, the atmospheric absorption of 1550 nm light is less than other wavelengths in the NIR[4].

The Starshot mission not only emphasizes reducing mass through nanotechnology but also harnesses advancements in arrayed lasers to project energy directionally across vast distances, optimizing propulsion efficiency. The high-power lasers for propelling the spacecraft are expected to operate at a single wavelength. As the sail accelerates to high speeds, Doppler red-shifting will alter the wavelength of the light relative to the sail. This necessitates that these ultra-thin reflectors remain highly reflective over the Doppler bandshift. However, there is an inverse relationship between the thickness of the reflectors and their reflectivity bandwidth: ultra-thin reflectors exhibit high reflectance over a narrow band, while thicker reflectors, which can increase the bandwidth, also add significant mass. This added mass can hinder acceleration. Thus, balancing thickness and broadband operation is a major challenge, which is evaluated using a figure of merit (FOM) that includes reflectivity, mass, and the Doppler shift of the sail.

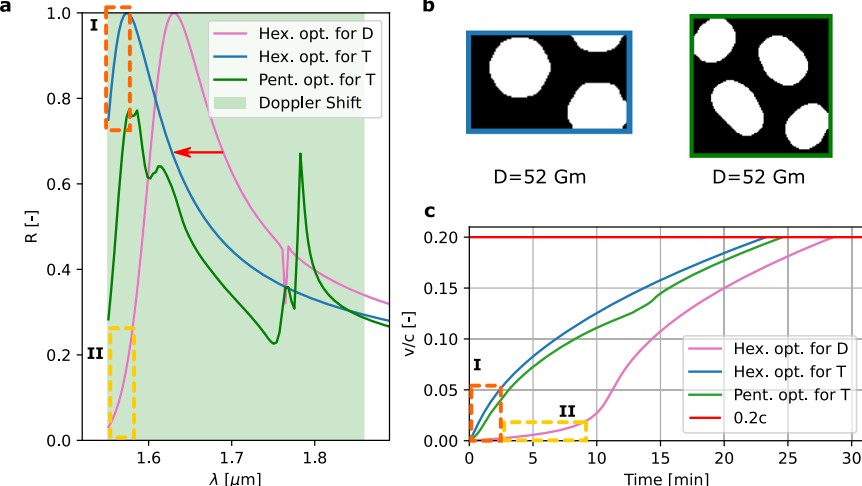

**Fig. 4 | Photonic crystal unitcell optimized for acceleration time. a** Reflectivity spectrum of two hexagonal PhCs. The red arrow indicates the shift of the reflectivity peak to the laser wavelength (i.e., to the left) when optimizing for acceleration time (T) instead of acceleration distance (D). **b** The final PhC designs optimized for T. The hexagonal PhC has an MFS of 517 nm and $A_f$ of 0.6 (blue). The pentagonal PhC has an MFS of 507 nm and $A_f$ of 0.63 (green). **c** The velocity of the PhC lightsail during acceleration compared with the speed of light. Regions I (orange) and II (yellow) indicate how the beginning of the reflectivity spectrum translates to the initial acceleration of the sail.

Initially, it is chosen to minimize the acceleration distance (D), i.e., the distance required to reach the final velocity of the lightsail, as the optimization objective. This quantity of interest is commonly used in lightsail design as it enforces a tradeoff between weight and broadband reflectivity[38]. Furthermore, it implicitly takes into account the laser's divergence limits[39]. The MFS imposed by the fabrication method must be included in the optimization to consider the fabrication cost. However, controlling the MFS explicitly is a challenging problem in the TO field[40] and becomes even more non-trivial for a neural network-based TO. Therefore, the MFS is generally controlled implicitly[41]. We extended the optimization with a simple approach of adding an Area fraction ($A_f$) as an extra optimization constraint[25] to control the MFS of the final design. The Area fraction can be calculated with $A_f = \frac{N_{cmat}}{N_{total}}$, where $N_{cmat}$ is the number of pixels with a refractive index of the PhC material and $N_{total}$ the total number of pixels.

For the final mission, the laser is presumed to emit a linear polarized plane wave. Optimizing a PhC lightsail for only one polarization direction results in parallel strings aligned with the polarization[6]. Thus, a precise and challenging alignment of the physical sail with the laser beam position and its polarization is required. Additionally, string-based PhCs are not practical for lightsail fabrication as they would stick together. Therefore, in this study, the sail is optimized for two orthogonal polarization angles $\phi = 0$ and $\phi = \frac{1}{2}\pi$ (i.e., rotation around the normal of the crystal plane) to obtain producible two-dimensional designs and promote a polarization direction invariant design, relaxing the alignment requirements. The optimization parameters used in the optimization are the relative permittivity of the pixels, the thickness and the period of the PhC. Additional information regarding the optimization and its formulation is described in the methods section.

**Computational results**

At first, the optimization was conducted without an area constraint ($A_f$), yielding designs with patterns following conventional square and hexagonal crystal lattices (Fig. 2b, c). However, these fall short of the MFS > 500 nm objective. Therefore, optimizations with a $A_f$ constraint from 40 to 70% were subsequently realized. Notably, larger $A_f$ led to a unique pentagonal lattice structure[42] (Fig. 2d). The performance of various designs is compared in Fig. 3a. The figure shows that an increasing $A_f$ correlates to an increase of MFS and, consequently, a decrease in performance (i.e., an increase of D).

The equation of motion of the lightsail[39] is solved directly from the reflectivity spectrum presented in Fig. 3b to obtain the sail velocity over its acceleration time (Fig. 3b), so the relation between the reflectivity spectrum and the performance can be studied. Intuitively, one would think that a design with a larger acceleration distance will also take more time to be accelerated. However, the key insight obtained from Fig 3c is that this is not the case. The pentagonal design with a higher D than the hexagonal design has a significantly lower acceleration time.

Additionally, the pentagonal design obtained by the neural TO method gives the non-trivial insight that a broadband reflector can be made with a two-dimensional PhC by designing it with multiple hole sizes and shapes, resulting in multiple resonance peaks. However, these peaks' total reflectivity is lower than that of a PhC designed with one fixed shape, thereby making the reflector more broad-band rather more than reflective. This allows a sail to be tuned to more wavelengths within the Doppler range, a quality not usually needed for conventional mirrors but critically important for lightsails. The supplementary information provides a more comprehensive analysis of the obtained designs, including the polarization dependence, acceleration distance, and time.

Notably, the launch cost is only determined by the acceleration time (T) (i.e., the time required to reach the final velocity of the lightsail), making it a significant performance parameter to consider. For example, when assuming ideal energy conversion to the laser and ideal momentum transfer to the sail, the time difference of 5 min between the pentagonal and hexagonal lattice, shown in Fig. 3c, can mean a difference in launch cost of 1.5 million euro with respect to a total launch cost of 9.3 million euro when calculating for a 10 GW/m² laser on a 10 m² sail and 0.185 euro/kWh[43] (average non-household energy price 2023). Considering the high throughput of launches required for Starshot missions, the costs of individual launches become of utmost importance.

Secondly, the lightsail is optimized to minimize T as the FOM because of the large impact of T on the launch cost. The formulation of the FOM can be found in the methods section. For this optimization, the obtained designs are the same as when optimizing for D. The best design from this optimization, which satisfies the MFS objective, is presented in Fig. 4b. Notably, the best design optimized for T follows a hexagonal lattice and has reduced the launch time by 6 minutes compared to the design optimized for D. This decrease in launch time

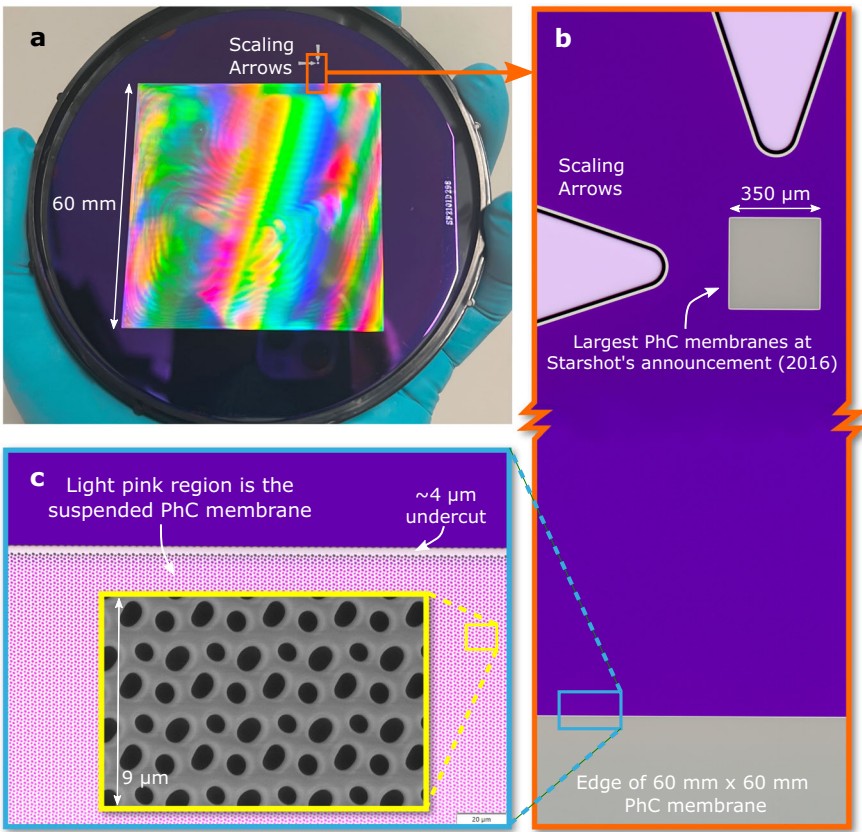

**Fig. 5 | High aspect ratio suspended photonic crystal membrane lightsail material. a** Photograph of a 100 mm wafer with a $60 \times 60$ mm², 200 nm thick suspended SiN PhC membrane, covered with a pentagonal pattern having a period of 3.0 μm. **b** Microscope image of two arrows etched into the substrate pointing towards a $350 \times 350$ μm² suspended PhC membrane. The bottom of the orange-framed inset shows the edge of the $60 \times 60$ mm² suspended membrane in the same magnification. The $350 \times 350$ μm² membrane puts the large membrane in perspective by showcasing the largest single-layer suspended PhC membranes at Starshot's announcement (2016)[16]. **c** 50x magnification of the edge of the membrane. One can see the repeating pattern covering the $60 \times 60$ mm² phononic crystal (PhC). The SiN is still attached to the silicon frame in the purple regions. The light pink indicates where the silicon has been removed under the PhC, leaving a suspended SiN PhC membrane. The yellow-framed inset shows a further zoom of the pentagonal lattice taken with a scanning electron microscope.

results in a cost reduction of approximately 2 million euro per launch (i.e., following the same calculation presented above).

However, there is a tradeoff between T and D. The decrease in T comes with the cost of an increased D. Figure 4a, c shows that the high reflectivity of the PhC for the initial wavelengths is responsible for the fast initial acceleration of the sail and, therefore, reaching its final velocity in less time. Alternatively, for a lightsail with low reflectivity at the initial wavelengths, the sail will only be accelerated slowly, wasting a lot of illumination time before it gets significantly accelerated. However, the reflectivity at the end of the spectrum is more important when optimizing for D because the lightsail should be accelerated fast when traveling at high speeds to prevent traveling excessive distance. So, when optimizing a PhC for only $D$, initial reflectivity is not prioritized and can result in a long acceleration time. Regarding the pentagonal lattice, optimizing for T or D did not change the performance significantly. However, when comparing the pentagonal and hexagonal designs optimized for T in Fig. 4b, it can be seen that both designs follow a similar path when accelerated, meaning that the designs are close together within the design space. This can indicate the neural TO finds the final solution in a basin where different designs have comparable outcomes for the FOM. Additional mission requirements can be included in the optimization to resolve this basin. Different requirements will call for other inherent properties of a PhC and determine the most suitable crystal structure for the lightsail application. For example, a notable difference between the pentagonal and hexagonal designs is that the $A_f$ for the pentagonal design is higher.

This property can be beneficial to aid radiative cooling and reduce stress concentration within the sail during its dynamic operation. In terms of fabrication, a larger $A_f$ means more material between holes, and more robust structures. In contrast, small $A_f$ would mean PhC designs characterized by small delicate wires of materials between holes which must survive the fabrication process of suspending the structures and subsequently undergo fast accelerations. The Area fraction will be a crucial parameter affecting several other important factors, including costs from MFS and manufacturability (i.e., the ability not to fracture too easily) and acceleration capabilities.

Transitioning to the broader context of existing literature, different two-dimensional PhC designs have been proposed before, with designs having a MFS between 125 and 260 nm, and D between 1.9 and 13 Gm[5,31,44–46]. However, meaningful comparisons with our findings pose a challenge due to the different mission parameters employed in previous studies. Factors such as variations in payload mass, laser power, and sail material limit the direct applicability of our optimized designs to those reported in the literature. Hence, newly proposed designs in this field should ideally utilize the same mission parameters. Nevertheless, it is worth noting that, despite lower D, the designs proposed in the literature are challenging or expensive to fabricate due to their intricate features and material choices. Additionally, there is a challenging tradeoff between preferable physical properties like broadband reflectivity, stability or cooling and the thickness of the PhC[6,8,30,32]. However, the extra mass can significantly increase the acceleration distance and time and thereby the cost.

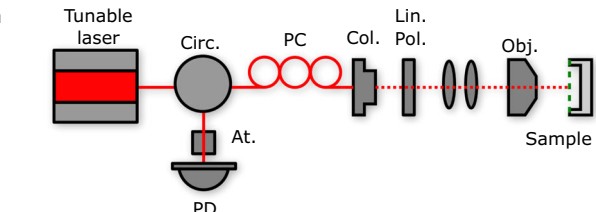

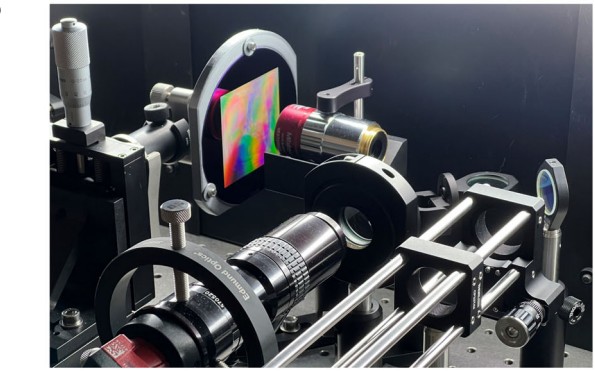

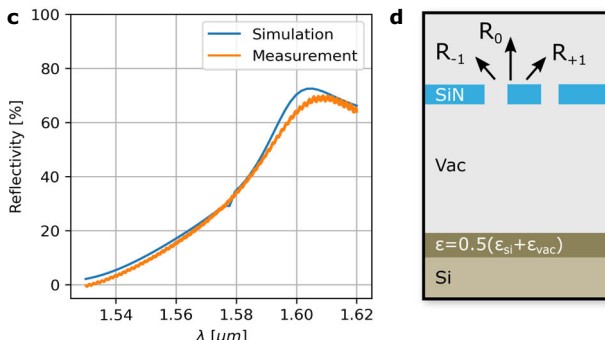

**Fig. 6 | Reflectivity spectrum validation. a** Experimental setup. Circ, circulator; PC, polarization controller; Col., collimator; Lin. Pol., linear polarizer; obj., objective; PD, photodetector. **b** 100 mm diameter wafer with $60 \times 60 \, mm^2$ suspended PhC membrane clamped in the measurement setup. **c** Simulations from the actual fabricated design obtained from the scanning electron microscope (blue), measurement (orange) with respect to a silver mirror reference. **d** Schematic representation of the simulation. A small layer with a relative permittivity of $(\epsilon_{si} + \epsilon_{vac})/2$[55] is introduced to represent the rough surface resulting from the undercut of the SiN membrane.

## Experimental results

For the reasons invoked previously, the pentagonal design presented in Fig. 3, was chosen to be fabricated in this work as a proof of concept. Fabricating a pentagonal lattice PhC membrane also illustrates the robustness of the fabrication method (elaborated in the Methods section). Figure 5 shows $60 \times 60 \, mm^2$ and a $350 \times 350 \, \mu m^2$ suspended single-layer PhC membrane. To illustrate the large scales of these suspended devices, we have etched millimeter-scale arrows pointing towards the smaller membrane. The smaller membrane represents the largest photonic crystals made at Starshot's announcement in 2016[16], highlighting a nearly 30,000-fold increase in surface area of SiN photonic crystal materials. Here, we are able to produce this reflective material at nearly 9000 times reduced cost per square meter. This demonstrates the progress in the scalability and aspect ratio achievable with our fabrication method and design methodologies that consider manufacturing yield. Remarkably, the device shown in Fig. 5 is one of the largest single-layer suspended PhC to date, having the highest aspect ratio (length/thickness ~$3 \times 10^5$) of any nanophotonic element and covered in about 1.5 billion nanoscale holes. To give an intuitive sense of this aspect-ratio, our 200 nm-thick photonic crystal

scaled up to a 1 mm thick glass sheet would extend for nearly $\frac{1}{3}$ km laterally, covered in ≈2.5 mm-diameter holes with ≈2.5 mm of glass between holes – an aspect ratio that is far beyond anything manufactured at macroscopic scales. At nanoscales where weight and forces scale differently due to low masses and small surface areas, unique high-aspect-ratio geometries become possible to produce.

Once suspended, the membrane is notably robust. The high tensile stress within the SiN membrane adds stability by keeping it taut, which prevents flapping or bending that could otherwise introduce additional stress points. This tension allows the membrane to withstand even air pressures during transport and handling, as long as direct contact with sharp or pointy objects is avoided.

A tunable laser (range: 1530–1620 nm) is used in the measurement setup (Fig. 6a and Methods) for obtaining a part of the reflectivity spectrum to validate the simulations. The range of the laser does not cover the full bandwidth of the PhC mirror. However, aligning the apparent peak/valley in the measured reflectivity spectrum with our simulations allows us to approximate the design's performance over the broader range despite the limited measurement range. The measurement and the simulations are shown in Fig. 6c. The measured value differs from the original spectrum due to the fabrication steps, like etching, which etch away some of the membrane's thickness during the undercut and enlarges the holes due to non-perfect selectivity. Therefore, the final shape of the PhC is retrieved via an electron microscope and used to obtain the expected reflectivity. Notably, the measurement performed is in good agreement with the simulation of the fabricated PhC.

Analysis of the final membranes revealed that the hole size of the PhC at the edge is approximately three percent larger than that of the center holes. This size difference causes a small shift in the reflectivity spectrum of 10–20 nm, likely because the etch rate in the middle is lower due to more etchant chemicals being available at the edge. Therefore, the middle, having more exposed silicon than the edges, consumes more $SF_6$ chemical (used to undercut our PhCs) and reduces the etch rate compared to the edge, which is adjacent to the substrate without holes and does not consume the chemical. This results in the membrane releasing first from the edges and then from the center, which is advantageous since we use cryogenic temperatures to improve the SiN/Si selectivity of the $SF_6$ etchant. Thus, the center remains well anchored thermally to the substrate during the release. The fabrication process can be optimized to counter the above-mentioned variations for an even better match with the optimized design. However, the suspended PhC membrane measurement is in reasonable agreement with the simulations. Additionally, the interference pattern, visible at the surface of the membrane (5a) shows the uneven etching underneath the $60 \times 60 \, mm^2$ PhC membrane. This pattern originates from the light interfering with the variable gap distance between the Si substrate and the SiN membrane. The supplementary information contains a study regarding the flatness of the PhC membrane.

## Discussion

High-aspect-ratio PhC reflectors, with subwavelength thickness and centimeter-scale dimensions, offer unique capabilities not achievable at smaller micron scales, as shown in Fig. 5b. Centimeter-scale photonic crystals can achieve higher reflectivity with thinner geometries because they do not require light to be focused down, which can severely reduce reflectivity from the ideal case of a plane wave incident on a PhC. In these larger-scale PhCs, the incident beam can interact with billions of nanoholes, similar to a plane wave on an infinitely sized PhC[9]. Their novel geometries open new possibilities for lightweight, compliant reflectors in dynamic applications like movable mirrors[47], imaging optics[48], as well as for acceleration to high speeds in space exploration.

This study presents the fabrication of the largest single-layer suspended photonic crystal (PhC) with the highest aspect ratio achieved for a nanophotonic element, marking an advancement for large-scale

PhC lightsails. Notably, we have achieved a 9000 times reduction in manufacturing costs, a critical breakthrough for the project's viability. This cost reduction stems from surpassing the MFS threshold set by diffraction, allowing the use of high-throughput photolithography for large wafer-scale production at significantly lower costs. We use the Area fraction of the photonic crystal as a way of optimizing for MFS, which is traditionally difficult in topology optimization.

Previous research has focused on optimizing acceleration performance, but this study directly addresses the critical costs of manufacturing, yield (i.e., lightsail breakage), and laser launching. The Starshot project's shotgun approach highlights that economic considerations are as crucial as scientific performance for mission success. The coupling of economics and performance will ultimately determine feasibility and can lead to non-intuitive design strategies.

The design process was conducted by neural topology optimization (neural TO). This method was found to be more robust than performing topology optimization without the neural network reparameterization trick, as it does not require artificial relaxation in the simulations to converge to optimum solutions. Traditional PhCs can be highly reflective at a single wavelength peak or more broadband by increasing thickness, which adds mass. Constrained to a single layer of SiN, neural TO discovered a basin of possible designs with similar performance and a novel periodic placement of holes: the pentagonal lattice. This lattice features several peaks of relatively lower reflectivity strategically spread over a broad wavelength range, optimizing to reduce acceleration time. This innovative approach demonstrates that a pentagonal lattice with multi-shaped periodic structures offers extra degrees of freedom, enabling the ability to tune beneficial trade-offs between reflectivity, broadband operation, and Area fraction.

A key insight is that the feasibility of a lightsail mission to Alpha Centauri will hinge on balancing manufacturing costs and performance, both linked to the PhC Area fraction. A high Area fraction reduces manufacturing costs and improves yield but hurts acceleration performance. Additionally, a low Area fraction enhances acceleration but increases manufacturing complexity and costs. Neural TO navigates this optimization landscape by balancing these demands.

Integrating the cost-saving measures discussed, the total savings per sail are substantial. By reducing manufacturing costs by 9000 times and optimizing launch costs by focusing on acceleration time, we estimate significant overall budget reductions approaching 25 million euro per lightsail. This continual focus on cutting costs is essential for using lightsails for space exploration.

Future research should explore multi-objective topology optimization, incorporating structural[49], thermal[10,31], and photonic stability[8,32,50,51] parameters to develop viable lightsails producible by cost-effective methods. Including realistic constraints, such as mass penalties for the lightsail's connection to the payload, will also be crucial. Additionally, the influence of absorption of the light within a PhC membrane needs to be studied to understand its behavior when illuminated with a high-power laser.

This study also demonstrates the potential of neural topology optimization to achieve innovative and economically viable lightsail designs, crucial for next-generation space exploration. We demonstrate that wafer-scale subwavelength-thickness reflectors can be produced in a truly scalable manner by optimizing both manufacturing costs and design. In principle, our techniques allow for the production of these low-mass, broadband reflectors at any wafer size currently available in the semiconductor industry (currently 400 mm diameter). Integrating economic and performance considerations will be pivotal for the feasibility and success of ambitious projects like the Starshot Initiative. While the trajectories for such a mission are ambitious, these goals initiate a new exploration of extreme light-matter interactions, leading to advancements in photonics, structural engineering, and materials science, and opening up a new regime in these fields.

## Methods

### Optimization formulation

The neural TO algorithm is divided into four sections: convolutional neural network (CNN), post-processing, functional analysis, and calculation of the figure of merit (FOM)[25]. Furthermore, the optimization consists of a forward and backward step. The forward step involves feeding a randomized vector $\beta$ into the CNN, which produces the image of the optimized structure (i.e., the discretized design space). This image is filtered, after which the performance parameters obtained in the functional analysis can be used to determine the FOM. In the backward step, the gradients with respect to the FOM are calculated for all the trainable variables of the CNN and the elements of $\beta$ so that the L-BFGS[52] optimizer can be used to update them at each new iteration. This procedure is repeated until the FOM reaches a pre-set relative tolerance or a maximum number of iterations. The performance of the neural TO approach is discussed in more detail in the supplementary information.

For the optimization of the PhC based lightsail, only the PhC unit cell is considered. In this optimization, a two-dimensional design space is discretized into a grid of $N \times N$ (square lattice), and $N \times \sqrt{3}N$ (hexagonal lattice) pixels, and these pixels' material properties can be continuously varied between the vacuum and the required material. Furthermore, the unit cell's period $\Lambda$ (i.e., the lattice vector) and the layer thickness $t$ are used as optimization parameters. A schematic overview of the optimization and the parameters is presented in Fig. 7.

The functional analysis is performed with Rigorous coupled-wave analysis because this semi-analytical method is computationally efficient in solving scattering problems for periodic structures with layers that are invariant in the direction normal to the periodicity[6].

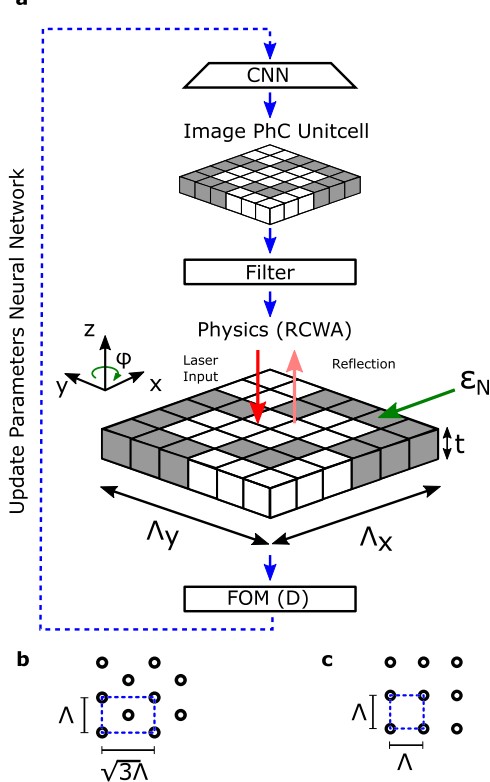

**Fig. 7 | Schematic of lightsail neural topology optimization. a** The lightsail is optimized for one layer with thickness $t$. The unit cell with the period of $\Lambda$ is optimized. The discretized voxels (N) of material have an assigned dielectric constant $\epsilon_N$. CNN, Convolutional neural network; RCWA, rigorous coupled-wave analysis; FOM, Figure of merit. The shape of the design space for a hexagonal (**b**) or square (**c**) lattice.

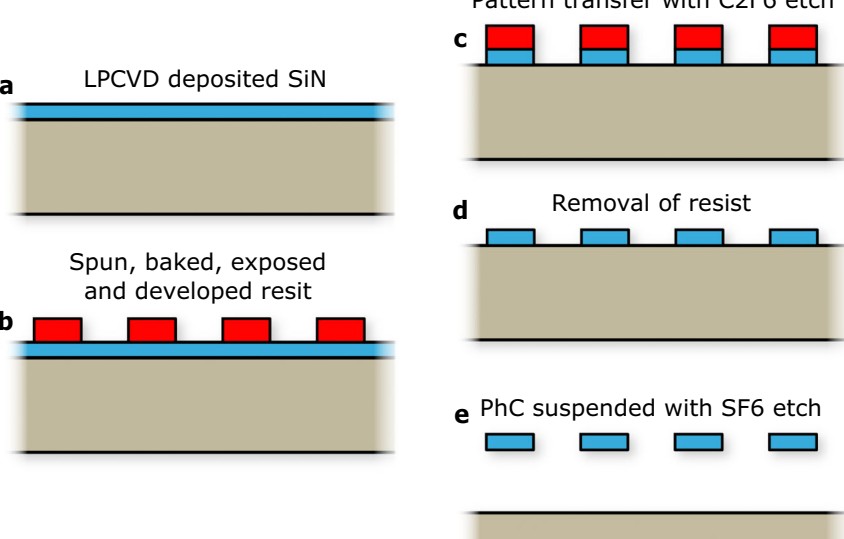

**Fig. 8 | Fabrication of cm-scale suspended photonic crystal membranes.** Schematic overview of the fabrication steps of the suspended PhC lightsail[54]. **a** SiN (blue) positioned on a Si wafer with low-pressure chemical vapor deposition (LPCVD).

**b** Patterning of the photoresist mask (red), **c** Directional $C_2F_6$ plasma etch, **d** Resist removal, **e** $SF_6$ undercut.

Minimizing $D$ is chosen as the first objective for the design optimization. The formulation for the acceleration distance is presented in Eq. (2).

$$D = \frac{c^3}{2I}(\rho_l + \rho_s) \int_0^{\beta_f} \frac{h(\beta)}{R[\lambda(\beta)]} \, d\beta \qquad (2)$$

In this equation, $D$ is the acceleration distance, $I$ is the intensity of the propulsion laser, $\rho_l$ and $\rho_s$ are the area densities of the lightsail and the satellite respectively, $\lambda$ is the wavelength of the propulsion laser, R is the reflection as a function of $\lambda$, and $h(\beta) = \beta/(1-\beta)^2 \sqrt{1-\beta^2}$, where $\beta$ is the velocity fraction with respect to the speed of light $\beta = v/c$. Due to the Doppler red-shift of the laser, the wavelength of the laser can be written as a function of the relative speed, $\lambda(\beta) = \lambda_0 \sqrt{(1+\beta)/(1-\beta)}$. When using a 1.55 μm laser, the bandwidth at which the sail will operate is from 1.55 μm to 1.86 μm. $\rho_l$ and $R$ are the geometry-dependent parameters.

Secondly, the lightsail is optimized for $T$[39]. The formulation of T is presented in Eq. (3).

$$T = \frac{m_t c^3}{2IA} \int_0^{\beta_f} \frac{\gamma(\beta)^3}{R[\lambda(\beta)]} \left(\frac{1+\beta}{1-\beta}\right) d\beta \qquad (3)$$

For this equation, $m_t$ and $A$ are the total mass and area of the sail respectively, $\gamma(\beta) = 1/\sqrt{1-\beta^2}$.

The optimization aims to minimize the FOM for a SiN lightsail following the 2016 Starshot parameters. The density of SiN is set to 3100 kg/m³[53], and its relative permittivity is 4[37]. The relative permittivity of the pixels is varied between 1 (vacuum) and 4 (SiN). The thickness ($t$) and period ($\Lambda$) are constrained to 0.01 μm ≤ $t$ ≤1 μm and 0.1 μm ≤ $\Lambda$ ≤ 7.2 μm respectively. The intensity $I$ of the laser beam is set to 10 GW/m² with a wavelength $\lambda_0$ of 1.55 μm, illuminating a sail area of 10 m². The laser is assumed to be a linear polarized plane wave, and the sail is optimized for two orthogonal polarization angles $\phi = 0$ and $\phi = \frac{1}{2}\pi$. The initial solution of the material distribution is random, and the initial solution of the thickness and period is set to 100 nm and $\lambda_0$ respectively. The design space is divided into a 100 × 100 and 100 × 172 pixel grid for a square and hexagonal lattice respectively.

## Nanofabrication of the PhC

The stringent mass requirements of the Starshot Initiative make hole-based photonic crystals (PhCs) inevitable. While multilayered and bilayer PhCs have a 100% fill factor (i.e., no holes), making them heavier and easier to fabricate, they are unsuitable due to their excessive mass. To approach the target mass of 1 g, we must employ single-layer PhCs with holes, achieving a fill factor of 40−70%. This design choice, although necessary for reducing weight, introduces fragility, as stress concentrations occur in the material between holes. Consequently, fabricating centimeter-scale nanophotonic reflectors that are both lightweight and robust poses significant challenges. Effective lithography of billions of holes must be achieved rapidly, and the high-aspect-ratio single-layer PhCs must be suspended with a single, stiction-free undercut using dry chemical etching.

Figure 8 displays the fabrication process of the suspended PhC lightsail, of which a similar process is described in the work of Shin et al.[54]. Initially, a 100 mm Si wafer is covered with 200 nm of silicon nitride using low-pressure chemical vapour deposition (LPCVD) to attain a pre-stress of 270 MPa. Next, the AZ ECI 3012 positive-tone resist is spin coated. Before the coating a HMDS and baking step is performed at 130 °C for 30 s and 60 s respectively. The resist is then spin coated at 6850 rpm to reach 1 μm thickness and soft baked at 95 °C for 150 s. An ASML PAS 5500/80 automatic wafer stepper is used to expose the resist with a 110 mJ/cm² dose, operating with chrome on quartz mask to stitch 5 × 5 mm² patterns together to a 60 × 60 mm² PhC. The development consists of a PEB step at 115 °C for 150 s, followed by single puddle development with MF322 for 60 s at 3000 rpm. At last, the wafer is hard-baked at 100 °C for 150 s. The resist mask enables the PhC pattern to be transferred using a 60 s directional inductively coupled plasma (ICP) to etch the SiN layer with $C_2F_6$. Finally, a 45 s fluorine-based $SF_6$ ICP etch is used at −120 °C to suspend the PhC membrane.

## Measurement setup for reflectivity PhC

The large-scale suspended PhC membrane is measured in the setup shown in Fig. 6a to obtain the actual reflectivity spectrum and to validate the simulations. The setup consists of a tunable laser that emits a laser beam from 1530 to 1620 nm. This laser beam passes through an optical fibre to a circulator, followed by a polarization controller (PC). The light then goes into free space through a collimator. The light passes through a linear polarization filter to ensure it

is linearly polarized. The beam size is then decreased and focused with a lens set and an objective. The reflected light follows the same path back and is diverted in the circulator to an attenuator and finally to the photodiode. The measurements are performed by first maximizing the signal retrieved from a silver mirror at 1580 nm, after which its reflectivity spectrum is measured, and this is repeated for the PhC membranes. The actual reflectivity could be obtained by normalizing the measurement of the PhC membrane with the known reflectivity of the silver mirror. Figure 6b shows the 100 mm diameter wafer within the measurement setup.

## Simulation of measurement

The PhC unit cell of the lightsail is simulated as a SiN membrane surrounded by a vacuum in the lightsail TO. However, the fabricated design is a suspended SiN membrane attached to a Si wafer. Therefore, to match the measurements with the simulations, the full system must be considered as presented in Fig. 6d. This figure shows the vacuum gap between the SiN membrane and the Si wafer, which is around 4 μm and can be deducted from the undercut at the edge of the membrane through optical microscopy. Additionally, a layer around 100 nm with a refractive index of $(\epsilon_{Si} + \epsilon_{vac})/2$[55] was added on top of the Si wafer, representing the roughness resulting from the undercut. This resulted in a qualitative match of the measurement, yet the total reflectivity did not match. This mismatch is because the setup is designed only to measure the normally reflected, zero-order diffracted light. However, the total reflectivity is used in the lightsail optimization to obtain the total momentum transfer in the normal direction of the sail. Consequently, after fitting the height of the gap, the roughness layer and when only considering the zero-order reflected light, the simulation is in good agreement with the measurements.

## Data availability

The data supporting the findings of this study are available in a 4TU database with the DOI identifier https://doi.org/10.4121/fa47f4e9-4c28-4b50-9615-b35216aacdfc.

## Code availability

The optimization code is being generalized into one software language and is extensively documented. To avoid having two repositories in different languages using the same method, the code used for this paper will be available upon request from the corresponding author, M.A.B.

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

## Acknowledgements

We want to thank Yufan Li, Peter Steeneken, Megha Khokhar, Mark Kalsbeek, Juan Lizarraga Lallana and Ata Keşkeklar for stimulating discussions. Additionally, we want to thank R. Tufan Erdogan for his support with our measurements. We would like to thank Shushu Qin for her invaluable contributions to the development of the code used in this study. Funded/Co-funded by the European Union (ERC, EARS, 101042855). Views and opinions expressed are however those of the author(s) only and do not necessarily reflect those of the European Union or the European Research Council. Neither the European Union nor the granting authority can be held responsible for them. R.A.N. and M.A.B. would like to acknowledge support from the Limitless Space Institute's I$^2$ Grant.

## Author contributions

L.N., M.A.B., and R.A.N. Designed the research; L.N. and S.Y. conducted the computational design. L.N. fabricated the PhC membrane with the support of F.S., H.A., and P.M.S. L.N. led the experiment with the support of M.H.J.J. L.N. analyzed the data; L.N., S.Y., M.H.J.J., M.A.B., and R.A.N. wrote the paper.

## Competing interests

The authors declare no competing interests.
