## [Peer Review File · Nature Communications]

REVIEWER COMMENTS

Reviewer #1 (Remarks to the Author):

This paper presents intriguing theoretical and experimental work on laser sail design and fabrication. The authors fabricated impressively large patterned silicon nitride membranes with promising optical properties and highlighted the importance of fabrication costs and acceleration time in light sail pattern design. Overall, the paper represents a significant advance in the field and would be suitable for publication in Nature Communications, provided the authors address the following:

- 1) The claim of a wrinkle-free membrane needs better characterization. For example, the interference pattern in Fig. 5a may indicate wrinkling. Is this interference due to changes in membrane angle or something else? The authors could measure out-of-plane displacement across the membrane using an optical profilometer or another tool to prove the absence of wrinkling.
- 2) Please explain the differences in the acceleration profile in Fig. 3b in more detail, particularly why the authors' algorithm results in a hexagonal design with such low reflectivity at 1.55 microns, causing extremely slow initial acceleration. Wouldn't a design scaled down by a factor of, say, 0.95 in the in-plane directions have substantially better reflectivity at 1.55 microns and therefore perform much better in terms of acceleration time?
- 3) Have the authors characterized absorption in their specific patterned film, rather than just referring to measurements of unpatterned films in Ref. 37? The edges in patterned films may introduce defects that increase absorption dramatically compared to unpatterned films.
- 4) Please comment on the robustness of the silicon nitride membranes. How many membranes have the authors made and tested? Can normal handling with tweezers or air pressure during out-of-plane movements break the membrane after it is suspended? Does the specific hole pattern (e.g., hexagonal vs pentagonal) affect robustness during handling?
- 5) Could the authors define the area fraction in the main text? I think I know what they mean (1-porosity) but it would be great to have an explicit definition.

Reviewer #2 (Remarks to the Author):

Please see the attached file for this reviewer's comment.

The authors investigate the structural optimization of two-dimensional photonic crystals made of silicon nitride for its lightsail application. The lightsail, which is assumed to be accelerated because of the radiation pressure of near-infrared laser light irradiated from the ground, is designed with a structure capable of reaching $0.2c$ in a short period of time. The sail has a stringent weight limit of approximately 1 gram for a size of 10 m^2 , and it is required to possess an MFS that can be realized using photolithography (Hg i-line) to enable fabrication within a realistic cost and timeframe. By utilizing an optimization method based on neural networks, several optimum solutions for photonic crystals with square, hexagonal, and pentagonal lattices have been demonstrated. Notably, the pentagonal lattice photonic crystal is discovered as a good solution with a large MFS of approximately 500 nm, while achieving a T of around 25 minutes. Experimentally, the reflection spectrum of the pentagonal photonic crystal fabricated using photolithography with a size of $60 \times 60 \text{ mm}^2$ is presented, showing a good agreement with theoretical calculations.

This reviewer thinks that this work includes outstanding achievements that they solved a complex structural optimization problem on photonic crystals for lightsail applications. Particularly, for MFS and T which have large contribution to the overall cost, the demonstration of the optimum structures for several photonic crystal lattices is a significant achievement. Moreover, the experimental validation of the reflection spectrum of the optimized pentagonal lattice structure is a noteworthy result, even though the measured wavelength range is limited. The content is a hot research topic and deemed suitable for publication in this journal, but some unclear points in the discussion warrant revisions.

Q1. Regarding the laser source

On page 5, the left column discusses the power consumption of the laser. What type of laser is assumed to be used? Will it be a laser array composed of commercial semiconductor lasers?

Q2. Regarding the thickness of the structure

The thickness of photonic crystals is fixed at 200 nm in this study. How was this thickness determined?

Q3. Differences in the T-optimal solutions between pentagonal and hexagonal lattices

In Figure 4(b), the T-value for the hexagonal lattice is lower than that of the pentagonal lattice. How about the MFS for these structures? Are both structures feasible for fabrication using photolithography (Hg i-line)? I did not fully understand which point of pentagonal lattice is superior to the hexagonal lattice when photonic crystals of both lattices are optimized for T. The discussion in the right column on page 5 argues that a higher Af is a superior aspect of the pentagonal lattice, but is this the only factor? If so, it would be beneficial to clarify the Af value of hexagonal and pentagonal photonic crystals optimized for T.

Q4. Regarding the measurement of the reflection spectrum

- The authors assume that the wavelength of the laser light irradiating the sail will shift from 1.55 μm to 1.85 μm due to the Doppler effect, but the wavelength range of the reflection spectrum shown in Figure 6(c) is 1.54 μm to 1.62 μm . Since the shape of the reflection spectrum is used to calculate T, I believe it would be important to show the spectrum for that range.
- Does the reflection spectrum in Figure 6(c) show the same shape for orthogonal incident polarizations?
- Is there any positional dependence in the reflection spectrum shown in Figure 6(c)? Generally, when fabricating large-area thin photonic crystals, spatial non-uniformity is expected due to warping and other fabrication imperfections.

Q5. Prospects for further expanding the sail size

When ultimately fabricating a 10m² sail, how do you envision the fabrication process? I believe it would be challenging to fabricate this size in a single batch using current photolithography equipment. Moreover, if multiple photonic crystals are connected instead of being fabricated in a single batch, are there any techniques available for connecting them without adding weight?

Response to Reviewers:
"Pentagonal Photonic Crystal Mirrors:
Scalable Lightsails with Enhanced Acceleration
via Neural Topology Optimization"

L. Norder, S. Yin, M. H. J. de Jong, F. Stallone, H. Aydogmus,
M. A. Bessa, R. A. Norte

November 2024

We sincerely thank the reviewers for their time, thoughtful questions, and positive perspectives on our research. Their feedback has been invaluable in enhancing the clarity and quality of our manuscript.

In the following document, we have addressed each comment point by point, with all changes to the manuscript marked in red for easy reference. We believe the manuscript is now significantly improved, and we are grateful for the opportunity to refine our work with the reviewers' guidance. Thank you once again for your careful consideration and enthusiasm for our research.

1 Comments by Reviewer 1

This paper presents intriguing theoretical and experimental work on laser sail design and fabrication. The authors fabricated impressively large patterned silicon nitride membranes with promising optical properties and highlighted the importance of fabrication costs and acceleration time in light sail pattern design. Overall, the paper represents a significant advance in the field and would be suitable for publication in Nature Communications, provided the authors address the following:

1.1 Question 1

The claim of a wrinkle-free membrane needs better characterization. For example, the interference pattern in Fig. 5a may indicate wrinkling. Is this interference due to changes in membrane angle or something else? The authors could measure out-of-plane displacement across the membrane using an optical profilometer or another tool to prove the absence of wrinkling.

Fig. R 1: **a**, Stitched microscopy image of the corner of the 60×60 PhC membrane. Shows a microscopy image where multiple images with $100 \times$ magnification are stitched together. The hatch pattern on the image is due to the uneven lighting of the microscope. In this figure, we do not see the interference pattern as we have an objective with a small depth of field, confirming that the membrane is not wrinkled. **b**, one image of the corner of the membrane with $500 \times$ magnification.

We appreciate this observation, as questions regarding membrane wrinkling often arise from photographs in Fig 5a/6b, and it provides an good opportunity to clarify our findings in the manuscript. The light interference pattern seen in photographs results not from membrane wrinkling but from reflections on the uneven substrate surface beneath the semi-transparent membrane. This pattern is due to the substrate’s inherent roughness and the slightly varying gap created by the membrane’s gas-based undercut.

Primary Evidence: To offer clear visual evidence, we first present microscope images in Fig. R1, which accurately represent membrane flatness. The observed color in these images is highly sensitive to wrinkles with even of tens of nanometers in height, which would appear as dark features if present. Unlike standard photography, microscopy allows high precision in focusing directly on the membrane, giving a true sense of color and flatness. The absence of dark wrinkles anywhere on the membrane (see images, Fig. R1) serves as the simplest and primary evidence that the membrane remains wrinkle-free due to high stresses in the silicon nitride.

Additional Evidence: Additionally, white light interferometric measurements in Fig. R2a-d provide a more detailed look. At low magnification ($2\times$), interference from substrate reflections is visible; however, at $20\times$ magnification (Fig. R2d), these reflections disappear, confirming that the interference pattern originates from the substrate rather than wrinkles in the membrane. This difference in measurements across magnifications further supports that the membrane itself is smooth and wrinkle-free.

Our deposition process induces a tensile stress of 270 MPa, which greatly contributes to membrane flatness. Although this may not initially sound significant, it is comparable to stress induced in a cross-section of duct tape if an

Fig. R 2: **a**, Photograph of the $60 \times 60 \text{ mm}^2$ PhC membrane. **b**, height map of full wafer measured with Bruker white light interferometer measured at $2\times$ magnification. **c**, height measurement over the white line shown in (a). **d**, height map of the green region, measured at $20\times$ magnification. **e**, height measurement over the white line performed at $20\times$ magnification. **f**, microscopy image of the corner of the membrane (yellow). **g**, zoom in on the corner of the measurement shown in (a)(blue). **h**, height measurement over the white line in the blue region.

average grand piano were suspended off of it. This level of tension ensures that the membrane remains extremely flat.

In response to this valuable feedback, we have clarified this explanation in the main text (line 537-544) and included relevant figures in the Supplementary Information.

1.2 Question 2

Please explain the differences in the acceleration profile in Fig. 3b in more detail, particularly why the authors' algorithm results in a hexagonal design with such low reflectivity at 1.55 microns, causing extremely slow initial acceleration. Wouldn't a design scaled down by a factor of, say, 0.95 in the in-plane directions have substantially better reflectivity at 1.55 microns and therefore perform much better in terms of acceleration time?

The reviewer is correct when stating that the acceleration time would improve by changing the design (pink line) of Fig. 3 in the main text. We want to show this improvement in acceleration time and the inherent tradeoff between acceleration time and distance in Fig. 4. However, we note that the results from Fig. 3 were obtained by optimizing for *acceleration distance*, instead of acceleration time.

Fig. 4a,c shows that the high reflectivity at the initial wavelengths for the

PhC optimized for acceleration time is responsible for the fast initial acceleration of the sail and, therefore, reaching its final velocity in less time. Alternatively, for a lightsail with low reflectivity at the initial wavelengths (and higher at final wavelengths), the sail will only be accelerated slowly, wasting a lot of illumination time before it gets significantly accelerated later. However, the reflectivity at the end of the spectrum is more important when optimizing for acceleration distance because the lightsail should be accelerated fast when traveling at high speeds to prevent traveling excessive distance during the acceleration phase. For example, one could have a lightsail that has low reflectivity at the beginning of the spectrum, therefore requiring a long time to be accelerated at the initial stage of the acceleration, and high reflectivity at the end of the spectrum and still having a low acceleration distance, because the sail traveled at low speed during the slow acceleration initial acceleration phase. So, when optimizing a PhC for only D , initial reflectivity is not prioritized and can result in a long acceleration time. The main text is modified to clarify this point (line 399-413).

1.3 Question 3

Have the authors characterized absorption in their specific patterned film rather than just referring to measurements of unpatterned films in Ref. 37? The edges in patterned films may introduce defects that increase absorption dramatically compared to unpatterned films.

We appreciate this insightful question, as absorption within photonic crystal (PhC) membranes is critical for lightsail applications. Although we have not directly characterized increase in absorption within our PhC membranes - as such an investigation would require intricate experimental setups beyond the scope of this study - we acknowledge that it is an important direction for future work. To our knowledge, this question has not been experimentally tackled in literature.

Silicon nitride (SiN) was chosen for its consistently low absorption relative to other materials with high tensile stress and a moderately high refractive index — an advantageous balance that has made it a leading candidate in lightsail literature. Generally, higher refractive index materials correlate with increased absorption, but SiN stands out with the reasonable refractive index and low-absorption. This has made it one of the most favorable materials available in fields like quantum optomechanics where ultra-low-absorption is crucial at the quantum level where a single phonon of heating via absorption are important to eliminate.

The question of additional absorption within PhCs specifically is complex and remains largely unaddressed in the literature. Unlike unpatterned films, PhCs become reflective by coupling incoming light into guided modes within the membrane, where these modes constructively and destructively interfere with incident and transmitted light, respectively. This process leads to enhanced light-membrane interaction, which in turn may increase absorption. Additionally, this guided mode interaction could introduce nonlinear effects, including

absorption, especially when high-powered lasers are used in lightsail applications. However, quantifying these effects remains an open challenge, as few studies have coupled PhCs with cavities to measure absorption directly; let alone compare to unpatterned membranes. One article we know of is by Chen et al. (2017), which tries to characterize the absorption of a PhC membrane within a cavity[1]. Although we included this reference before, we have now included this reference into the manuscript when discussing absorption.

Selecting a material with minimal inherent absorption, like SiN, is essential, as the PhC pattern will likely increase this baseline absorption, though the precise increase remains to be studied anywhere in the literature. Measuring such difference would be a great advance in itself that we hope to tackle in the future.

Our large-scale membranes could offer a unique platform for exploring this topic further. They allow for true plane-wave coupling into guided modes and facilitate studies on how thermal heating scales with lateral clamping—a valuable, alternative approach to cavity-based absorption measurements. We have added this point to the discussion and conclusion (line 626-629), as we believe our system could provide meaningful insights into this area.

1.4 Question 4

Please comment on the robustness of the silicon nitride membranes. How many membranes have the authors made and tested? Can normal handling with tweezers or air pressure during out-of-plane movements break the membrane after it is suspended? Does the specific hole pattern (e.g., hexagonal vs pentagonal) affect robustness during handling?

Membrane robustness has indeed been a critical focus in our work. Silicon nitride (SiN) is an exceptionally strong material, with an ultimate yield strength around 5 GPa, which is advantageous for ensuring the durability of large, suspended structures like those we employ. We found that, while robustness is high once the membranes are fully released, the most delicate stages are during fabrication, particularly at points where the membrane remains in contact with silicon before undercutting is complete. Sharp contact points can induce significant stress concentrations, potentially leading to fractures during this stage.

In response to these challenges, we developed optimized fabrication processes that significantly improved membrane yield. One key insight was to release the membrane as quickly as possible, minimizing attachment points that could become stress concentrators. Traditional fabrication using electron beam lithography often required mm-scale mainfields, where stitching errors of holes could compromise membrane integrity. In contrast, our approach with stepper photolithography enables nearly perfect stitching over large areas, allowing for faster, more consistent undercutting without introducing weak points. This technique has greatly increased our fabrication yield: for the last batch used in this work, all membranes fabricated from four wafers survived intact.

Fig. R 3: Microscopy image of $250 \times 250 \mu\text{m}^2$ PhC membrane on the tip of a tweezer

Once suspended, the membranes are notably robust regardless of hexagonal or pentagonal lattices. The high tensile stress within the SiN membrane adds stability by keeping it taut, which prevents flapping or bending that could otherwise introduce additional stress points. This tension allows the membrane to withstand even air pressures during transport and handling, as long as direct contact with sharp or pointy objects is avoided. Smaller PhC membranes, shown in Fig. R3, can even be held with tweezers without damage, demonstrating their resilience under controlled handling. For the lightsail application, the distributed load required is minimal. Using a $1 \text{ GW}/\text{m}^2$ laser, the resulting radiation pressure is around $6.7 \text{ N}/\text{m}^2$. The stress at the interface between the membrane and the substrate will be the highest. So, this pressure needs to be translated to an inplane pressure. The circumference of a 10 m^2 membrane is around 35 m and, together with the 200 nm thickness, this results in an area of $7 \times 10^{-6} \text{ m}^2$ over which the 67 N force needs to be distributed. This yields a pressure of 10 MPa, well below the material's yield or tensile strength. Thus, while the membranes are sensitive at their clamping points, as a whole they are highly stable under even pressures. Once suspended, they can be transported with ease from the cleanroom to the lab, and the added tension even enables movement without damage, as the tension prevents them from failing like an unstressed membrane.

To clarify these points, we have added a discussion of membrane robustness, including the impact of fabrication improvements, to the main text (line 36-39 + 492-498).

1.5 Question 5

Could the authors define the area fraction in the main text? I think I know what they mean (1-porosity) but it would be great to have an explicit definition.

Thank you. We added a clear definition in the main text (line 154-155 + 312-315). For the designs made in this work, we define the area fraction (A_f) as the fraction of the total area occupied by material. We call it the area fraction because we refer to a two-dimensional design. The definition for A_f is presented in Eq. (1), where N_{evac} is the number of pixels with a refractive index of vacuum and N_{total} the total number of pixels.

$$A_f = \frac{N_{\text{evac}}}{N_{\text{total}}} \quad (1)$$

2 Comments from Reviewer 2

The authors investigate the structural optimization of two-dimensional photonic crystals made of silicon nitride for its lightsail application. The lightsail, which is assumed to be accelerated because of the radiation pressure of near-infrared laser light irradiated from the ground, is designed with a structure capable of reaching $0.2c$ in a short period of time. The sail has a stringent weight limit of approximately 1 gram for a size of 10 m^2 , and it is required to possess an MFS that can be realized using photolithography (Hg i-line) to enable fabrication within a realistic cost and timeframe. By utilizing an optimization method based on neural networks, several optimum solutions for photonic crystals with square, hexagonal, and pentagonal lattices have been demonstrated. Notably, the pentagonal lattice photonic crystal is discovered as a good solution with a large MFS of approximately 500 nm, while achieving a T of around 25 minutes. Experimentally, the reflection spectrum of the pentagonal photonic crystal fabricated using photolithography with a size of $60 \times 60 \text{ mm}^2$ is presented, showing a good agreement with theoretical calculations. This reviewer thinks that this work includes outstanding achievements that they solved a complex structural optimization problem on photonic crystals for lightsail applications. Particularly, for MFS and T which have large contribution to the overall cost, the demonstration of the optimum structures for several photonic crystal lattices is a significant achievement. Moreover, the experimental validation of the reflection spectrum of the optimized pentagonal lattice structure is a noteworthy result, even though the measured wavelength range is limited. The content is a hot research topic and deemed suitable for publication in this journal, but some unclear points in the discussion warrant revisions

2.1 Question 1 - Regarding the laser source

On page 5, the left column discusses the power consumption of the laser. What type of laser is assumed to be used? Will it be a laser array composed of commercial semiconductor lasers?

While our work does not focus on laser technology, we recognize that several viable laser systems have been proposed for lightsail applications. For example,

Philip Lubin, a key advisor to the Starshot mission, has suggested a laser-phased array configuration that uses a large number of moderate-power laser amplifiers arranged in a master oscillator power amplifier (MOPA) setup with Yb amplifiers [2]. This approach eliminates the need for a single large laser source, providing a scalable solution with potential for efficient power delivery.

For our purposes, we assume a laser system capable of delivering the necessary output power and primarily consider the laser’s power requirements to estimate launch costs. For clarity, we have added this reference to the main text (line 11).

2.2 Question 2 - Regarding the thickness of the structure

The thickness of photonic crystals is fixed at 200 nm in this study. How was this thickness determined?

We understand how this detail could be missed, as this was only briefly mentioned in the Methods section and in the Supporting Information (Section E). In fact, we did consider thickness variations (see Fig. S5 in Supporting Information). In short, the optimization of the PhC membrane considered the following optimization parameters: relative permittivity at each pixel is varied between 1 (vacuum) and 4 (SiN); thickness (t) and period (Λ) are constrained to $0.01 \mu\text{m} \leq t \leq 1 \mu\text{m}$ and $0.1 \mu\text{m} \leq \Lambda \leq 7.2 \mu\text{m}$ respectively. We highlighted this in the methods section of the main text. The final design that we chose to fabricate had an optimized membrane thickness of 200 nm, which the optimizer found to be the optimal thickness for this specific design. We added a clear reference to the methods section in the main text (line 324).

2.3 Question 3 - Differences in the T-optimal solutions between pentagonal and hexagonal lattices

In Figure 4(b), the T-value for the hexagonal lattice is lower than that of the pentagonal lattice. How about the MFS for these structures? Are both structures feasible for fabrication using photolithography (Hg i-line)? I did not fully understand which point of pentagonal lattice is superior to the hexagonal lattice when photonic crystals of both lattices are optimized for T. The discussion in the right column on page 5 argues that a higher A_f is a superior aspect of the pentagonal lattice, but is this the only factor? If so, it would be beneficial to clarify the A_f value of hexagonal and pentagonal photonic crystals optimized for T.

The PhC designs optimized for acceleration time (T) in Fig. 4(b) have the following properties: the hexagonal PhC has a minimum feature size (MFS) of 517 nm and an area fraction (A_f) of 0.6, while the pentagonal PhC has an MFS of 507 nm and an A_f of 0.63. Both MFS values exceed 500 nm, making them feasible for fabrication using Hg i-line photolithography. We have added these A_f values to the Fig. 4 caption for clarity.

From an optimization perspective, the hexagonal and pentagonal PhCs perform similarly, indicating that the optimizer finds designs in a local basin within the solution space. This solution landscape could shift based on additional mission requirements, potentially revealing a clear optimal design.

However, the pentagonal lattice offers distinct advantages when considering multiple mission requirements, as it provides additional degrees of freedom. For instance, the hexagonal lattice is defined only by the hole radius and lattice constant, while the pentagonal lattice allows further tuning, such as by adjusting the hole shape and arrangement. This flexibility enables us to achieve two resonance peaks rather than one, allowing beneficial trade-offs between reflectivity, broadband operation, and area fraction. Additionally, the higher A_f for the pentagonal design may offer structural and thermal benefits, as noted in the main text.

Our method demonstrates the potential for discovering new types of photonic crystals that go beyond traditional square and hexagonal lattices, providing tunability not typically available in conventional designs. As future mission requirements become clearer, we anticipate the need to address factors like sail stability on the beam, thermal conduction, and alignment resilience. Unconventional lattices such as the pentagonal design could offer unique capabilities for meeting these complex demands, and we believe our approach opens up a promising path forward in the field.

2.4 Question 4 - Regarding the measurement of the reflection spectrum

1. *The authors assume that the wavelength of the laser light irradiating the sail will shift from 1.55 μm to 1.85 μm due to the Doppler effect, but the wavelength range of the reflection spectrum shown in Figure 6(c) is 1.54 μm to 1.62 μm . Since the shape of the reflection spectrum is used to calculate T , I believe it would be important to show the spectrum for that range.*

We agree, but unfortunately, we can only measure the reflection spectrum of the fabricated sample over the range from 1.54 μm to 1.64 μm because there are no lasers with the full bandwidth available to us. The best solution we could find within the practical constraints we have is to align the apparent peak/valley in the measured reflectivity spectrum with our simulations. The measured data falls within reasonable agreement given these constraints. This alignment allows us to approximate the performance of the design over the broader range with confidence, despite the limited measurement range. To make this clear, We have added this to the main text in line 502-507.

2. *Does the reflection spectrum in Figure 6(c) show the same shape for orthogonal incident polarizations?*

Fig. R 4: Polarization dependency of the final optimized pentagonal and hexagonal PhC design. **a** polarization orientation of incident plane wave. The polarization dependency of the reflectivity spectrum for hexagonal (**b**) and pentagonal (**c**) design.

For the figure of merit within our optimization, we calculate the acceleration distance or time for two orthogonal polarisation directions of the incident plane wave. This is performed such that the optimization is more robust to a possible misalignment of the sail with the laser, and to promote manufacturable designs without a preferential direction for the PhC (for example, some papers in the literature find "string based" structures that cannot be fabricated). However, the final PhC's performance depends on the polarization. In the supplementary information, Fig S1(c) shows the polarization dependency of multiple PhC designs obtained in the optimization for acceleration distance. With this figure, we wanted to show that the polarisation dependency cannot be ignored in the final sail design. Fig. R4 shows the polarization dependence of the designs from Fig. 4. From this figure, it can be seen that the reflectivity spectrum is slightly different for the different polarization directions. Therefore, the reflectivity spectrum presented in Fig. 6(c) will result in a slightly different spectrum when measured with an orthogonal polarization. Fig. R4 is added to the supplementary information.

3. *Is there any positional dependence in the reflection spectrum shown in Figure 6(c)? Generally, when fabricating large-area thin photonic crystals, spatial nonuniformity is expected due to warping and other fabrication imperfections.*

When measuring across the membrane from center to edge, we do observe slight variations in the reflection spectrum. Figure R5 shows four measurements taken at different points from the center to the edge of the membrane. By fitting these measurements to our model, we attribute the main cause of this variation to a slight thickness change across the

Fig. R 5: Reflectivity spectrum measured (blue) from the center to the edge of the sample together with the simulation (orange) of the 60 mm \times 60 mm suspended PhC membrane. For each measurement, the fitting parameters are presented

membrane, likely due to a position-dependent etch rate.

Further analysis under a Scanning Electron Microscope revealed that the PhC hole sizes vary by up to 3% across the membrane, likely due to differences in the etchant concentration across the sample. In the fabrication process, the center of the membrane consumes more SF_6 etchant, reducing the etch rate there compared to the edges, where less etchant is consumed. As a result, the membrane tends to release from the edges first and then from the center. This staggered release, which occurs at cryogenic temperatures to improve SiN/Si selectivity, helps keep the center thermally anchored to the substrate, reducing warping and maintaining structural stability during release. Figure R5 has been added to the supplementary information.

2.5 Question 5 - Prospects for further expanding the sail size

When ultimately fabricating a 10 m² sail, how do you envision the fabrication process? I believe it would be challenging to fabricate this size in a single batch using current photolithography equipment. Moreover, if multiple pho-

tonic crystals are connected instead of being fabricated in a single batch, are there any techniques available for connecting them without adding weight.

Indeed, the $3.6 \times 10^{-3} \text{ m}^2$ PhC membrane we fabricated is still far from the envisioned 10 m^2 sail. We agree that the two main approaches are to either fabricate a single large PhC membrane or connect multiple sail elements without adding significant weight. Given the long-term nature of the Starshot mission — akin to the multi-decade development of projects like LIGO—realizing such large-scale sails will likely require major advances in nanofabrication, particularly in scaling up wafer sizes.

Since the 1970s, wafer sizes have progressed from 100 mm to today’s 450 mm, and similar advancements could support the scaling of lightsail technology. A key point of our work is that our fabrication technique can adapt to any wafer size that becomes standard in the semiconductor industry, making it possible for lightsail fabrication to follow future developments in nanofabrication tools, processes, and sizes. Interestingly, as equipment scales up for larger wafers, the precision and consistency of etching and deposition processes often improve, which could enhance the feasibility of larger PhC structures.

However, whether such scaling will occur depends largely on economic incentives within the industry, as the economics of lightsail production are inherently linked to its unique geometry and requirements. Given the robustness of our photonic crystal materials, an alternative approach could be to overlap the edges of large PhC segments, allowing them to adhere through van der Waals forces. Although these overlapped regions would not reflect optimally, they would constitute only a small fraction of the sail’s total area and likely have minimal impact on overall performance.

It’s worth noting that just eight years ago, the largest PhCs were limited to 350×350 microns. Our work demonstrates a four-order-of-magnitude increase, and with continued advances, even larger scales may be possible. Fig. 5 in our main text illustrates that our current method can scale in line with future wafer sizes, offering a practical path forward for large-scale lightsail fabrication.

References

- [1] X. Chen, C. Chardin, K. Makles, C. Caër, S. Chua, R. Braive, I. Robert-Philip, T. Briant, P.-F. Cohadon, A. Heidmann, T. Jacqmin, and S. Deléglise. High-finesse fabry-perot cavities with bidimensional si_3n_4 photonic-crystal slabs. *Light: Science & Applications*, 6(1):e16190–e16190, 2017.
- [2] P. Lubin, A. N. Cohen, P. Meinhold, P. Srinivasan, N. Rupert, and P. Krogen. 7 - large-scale directed energythis book has a companion website hosting complementary materials. visit this url to access it: <https://www.elsevier.com/books-and-journals/book-companion/9780443159039>.: The path to radical transformation in propulsion. In C. Phipps, editor, *Laser Propulsion in Space*, Aerospace Engineering, pages 205–225. Elsevier, 2024.

REVIEWERS' COMMENTS

Reviewer #1 (Remarks to the Author):

The authors have largely addressed my previous comments so I am happy to recommend the paper for acceptance. The only concern I have is with the definition of area fraction. On line 154, the authors define it as fraction of the area occupied by the material. However, on line 313 they seem to define it as the area fraction occupied by vacuum. The authors need to resolve or clarify this apparent discrepancy.

Reviewer #2 (Remarks to the Author):

Thank you for addressing my questions.

Your responses have provided a clear understanding of the potential of the pentagonal photonic crystal. I believe that the high design flexibility of the new photonic crystal could be useful for satisfying stringent constraints that are required in light sail application. Additionally, I now have a much better understanding of the challenges involved in fabricating large-area photonic crystals.

The authors' work offers valuable insights and challenges for fabricating large-area photonic crystals. This provides a new research topic in the field of nanophotonics, and could stimulate the research community.

All of my questions have been answered thoroughly, and the manuscript has been appropriately revised. I recommend this manuscript for publication.